# Twenty-Week Dietary Supplementation with Beeswax Alcohol (BWA; Raydel^®^) Ameliorates High-Cholesterol-Induced Long-Term Dyslipidemia and Organ Damage in Hyperlipidemic Zebrafish in a Dose-Dependent Manner: A Comparative Analysis Between BWA and Coenzyme Q_10_

**DOI:** 10.3390/ph17111434

**Published:** 2024-10-26

**Authors:** Kyung-Hyun Cho, Ashutosh Bahuguna, Yunki Lee, Sang Hyuk Lee, Ji-Eun Kim

**Affiliations:** Raydel Research Institute, Medical Innovation Complex, Daegu 41061, Republic of Korea

**Keywords:** dyslipidemia, fatty liver, high-density lipoprotein (HDL), inflammation, senescence, zebrafish

## Abstract

Background/Objectives: Beeswax alcohol (BWA; Raydel^®^) is a blend of six long-chain aliphatic alcohols extracted from honeybee wax and is well known for its diverse functionality and health benefits. Herein, the efficacy of a BWA dietary intervention for 20 weeks was assessed to ameliorate high-cholesterol diet (HCD)-induced dyslipidemia and adverse effects on the vital organs of adult zebrafish. Methods: Adult zebrafish were fed different high-cholesterol diets (HCDs; final concentration of 4%, *w*/*w*) supplemented with BWA (final concentrations of 0.1%, 0.5% and 1.0%, *w*/*w*) or CoQ_10_ (final concentration of 1.0%). Following 20 weeks of supplementation, blood and different organs (liver, kidney, testes and ovaries) were collected, and biochemical, histological and immunohistochemical analyses were performed. Results: The results demonstrate a dose-dependent effect of BWA of mitigating HCD-induced mortality in zebrafish over the 20-week supplementation period, which was noticeably better than the effect exerted by coenzyme Q_10_ (CoQ_10_). Consistently, a dose-dependent effect of BWA consumption of curtailing HCD-induced total cholesterol (TC) and triglyceride (TG) levels and increasing high-density-lipoprotein cholesterol (HDL-C) levels was noticed. Compared with CoQ_10_ (final concentration of 1.0%, *w*/*w*), BWA (final concentration of 1.0%, *w*/*w*) displayed a significantly better effect of mitigating HCD-induced dyslipidemia, as evidenced by 1.2-fold (*p* < 0.05) and 2.0-fold (*p* < 0.05) lower TC and TG levels and 2.4-fold (*p* < 0.01) higher HDL-C levels. The histological analysis revealed substantial prevention of fatty liver changes, reactive oxygen species (ROS) generation, cellular senescence and interleukin (IL)-6 production in the hepatic tissue of BWA zebrafish, which was significantly better than the effect exerted by CoQ_10_. Consistently, compared with CoQ_10_, significant 25% (*p* < 0.05) and 35% (*p* < 0.01) reductions in the HCD-induced elevated levels of the hepatic function biomarkers aspartate aminotransferase and alanine aminotransferase was observed in the BWA group. Likewise, BWA consumption efficiently ameliorated HCD-induced kidney, ovary and testis damage by inhibiting ROS generation, cellular senescence and lipid accumulation. Conclusion: Supplementation with BWA demonstrated higher therapeutic potential than that with CoQ_10_ to prevent dyslipidemia and organ damage associated with long-term consumption of HCDs.

## 1. Introduction

Beeswax is a well-known natural ingredient produced by different species of honeybees and has a varied beneficial function [1]. The majority of commercially used beeswax is obtained from the honeybee species *Apis mellifera* and *Apis cerana* [2] and contains ~67% esters, ~16% hydrocarbons and ~12% free fatty acids, along with ~1% free alcohols. Beeswax alcohol (BWA), isolated from beeswax, consists of a blend of long-chain aliphatic alcohols (LCAAs) of C24 to C34 and is renowned for its potent antioxidant and anti-inflammatory activities [3,4,5]. Recently, the impact of BWA on the prevention of low-density-lipoprotein (LDL) oxidation and the functionality enhancement of high-density lipoproteins (HDLs) via augmenting HDL-associated paraoxonase (PON)-1 activity has been described [6]. Additionally, BWA substantially elevates high-density-lipoprotein cholesterol (HDL-C) levels and inhibits hepatic inflammation, leading to zebrafish protection against the stress induced by ethanol [7]. BWA entrapped in reconstituted HDL (rHDL) prevents HDL glycation and rescues zebrafish embryos from carboxymethyllysine (CML)-induced toxicity [8]. While numerous benefits have been identified, the significant impact of BWA on gastric health has been particularly highlighted in various preclinical and clinical studies [9,10]. In a preclinical study in rats, BWA supplementation was found effective against gastric ulceration induced by various stimulants without interfering with gastric acidity [10]. BWA’s impacts on the quality and quantity of gastric mucosa, reduced neutrophil infiltration and improved antioxidant status in gastric mucosa are among the key events behind the gastro-protective role of BWA [10]. BWA’s antioxidant effect is not confined solely to the gastric mucosa, as it also manifests as substantial modulatory effects on the antioxidant enzymes in the liver and brain [10,11], suggesting the broad applicability of BWA. In human subjects, BWA (100 mg/day) oral consumption for two months positively modulated regurgitation, bloating, sucking and flatulence scores [10]. Along with gastroprotection, BWA displayed a substantial effect on joint health on account of a noteworthy inhibitory impact on proinflammatory leukotriene B4 (LTB4), the main culprit behind the generation of superoxide and proinflammatory cytokines [10]. Based on the above-mentioned effects, the Korean Food and Drug Administration (KFDA) has approved the BWA as a functional food ingredient for gastric and joint health [12].

Coenzyme Q_10_ (CoQ_10_) is a lipid-soluble antioxidant synthesized in the human body and has been widely used as a dietary supplement. CoQ_10_ has diverse functions in the body, including antioxidant, anti-inflammatory and obligatory components for energy production in mitochondria [13]. Under normal and healthy conditions, intrinsic CoQ_10_ is sufficient in the intracellular and blood systems; however, in certain pathological conditions, such as degenerative muscle and cardiovascular diseases [14,15], oxidative stress and ageing can deplete intrinsic CoQ_10_, which thus needs to be supplemented externally to balance its level [16]. Nonetheless, the daily dosage range of CoQ_10_ is too wide, 30–600 mg (moderate), up to 600~3000 mg (high), depending on the requirement for curing target disorders [17]. Besides its noteworthy antioxidant role, the effect of CoQ_10_ on the lipid profile has not been observed sufficiently in animal experiments and human studies. A meta-analysis of randomized controlled trials revealed that supplementation with CoQ_10_ at around 100~300 mg for 4~48 weeks in patients with coronary artery disease (CAD) was effective in lowering total cholesterol (TC) and increasing HDL-C; however, it was not effective in improving triglyceride (TG) and low-density-lipoprotein cholesterol (LDL-C) [18]. A randomized clinical trial with type 2 diabetic patients (n = 20) showed that supplementation with CoQ_10_ at 150 mg/day for 12 weeks decreased TG but had no effect on TC. Intriguingly, a significantly negative effect of CoQ_10_ supplementation on the increase in HDL-C and reduction in LDL-C was observed [19]. Another meta-analysis on CoQ_10_ supplementation in adults revealed that a decrease in TC, LDL-C and TG levels and an increase in HDL-C levels were achieved at 400~500 mg/day, suggesting that high-dose CoQ_10_ might be associated with an improvement in the lipid profile [20]. In a comparative in vitro study, CoQ_10_ displayed a much inferior effect compared with BWA in preventing oxidative damage by LDL and was found ineffective in augmenting the functionality of HDL by modulating the activity of PON-1.

Numerous studies on BWA report on its dynamic functionality examined in vitro [6], in vivo [7] and in clinical settings [6,10]. However, studies on BWA dietary consumption’s impact on high-cholesterol diet-induced dyslipidemia and associated oxidative stress and senescence in different organs remain lacking. To address this, the current study was designed to evaluate the comparative influence of 20 weeks of dietary intervention with BWA and CoQ_10_ under the influence of a high-cholesterol diet on the plasma lipid profile, senescence and functionality of the liver, kidney and reproductive organs of hyperlipidemic zebrafish. Given that BWA is not soluble in water, for the comparative assessment, we selected lipid-soluble CoQ_10_ (as a control), which has been described as having several beneficial effects, including a substantial effect on the lipid profile [21].

## 2. Results

### 2.1. Change in Survivability and Body Weight

The results documented in Figure 1 demonstrate the survival probability during 20 weeks of consumption of the defined diets, which varied from 0.88 to 0.96 across the different groups. The HCD group exhibited the lowest survival probability (0.88), whereas the highest survival probability (0.96) was observed in the 0.1% and 1.0% BWA groups. Despite the slight difference in survival probability, the Kaplan–Meier survival analysis followed by a log-rank test revealed a non-significant difference (log-rank: χ^2^ = 4.891, *p* = 0.229) in zebrafish survivability among the different groups.

Figure 1B and Table 1 display the body weight changes in the different groups following 20 weeks of consumption of the different diets. At week 20, the 0.1% BWA and 0.5% BWA groups showed 13~14% lower body weight, and the 1.0% BWA group and 1.0% coenzyme Q_10_ (CoQ_10_) group showed 5~6% lower body weight than the HCD-only group; however, the changes in body weight were statistically non-significant (*p* > 0.05).

### 2.2. Change in Lipid Profiles

As shown in Figure 2A,B, the highest level of total cholesterol (TC; 584 mg/dL) and triglyceride (TG; 651 mg/dL) were detected in the HCD-only group following 20 weeks of consumption. Co-consumption of BWA efficiently altered the HCD-induced high levels of TC in a dose-dependent manner. A significant reduction of 43.2% (*p* < 0.001) in TC was quantified in the 1.0% BWA group compared with the HCD-only group, while the 1.0% coenzyme Q_10_ (CoQ_10_) group showed 27.7% lower (*p* < 0.01) TC levels than the HCD-only group. Importantly, the 1.0% BWA group exhibited 21.5% lower (*p* < 0.05) TC levels than the 1.0% CoQ_10_ group, suggesting that BWA showed higher efficacy in treating hypercholesterolemia induced by HCD supplementation.

Also, the co-consumption of BWA exhibited a substantial effect of lowering plasma TG levels in a dose-dependent manner, as evidenced by 31.9% (*p* < 0.01) and 62.2% lower (*p* < 0.001) TG levels detected in the 0.5% BWA and 1.0% BWA groups, respectively, compared with the HCD-only group (Figure 2B). Likewise, the 1.0% CoQ_10_ group also displayed a significant reduction of 23.1% (*p* < 0.05) in TG levels compared with the HCD-only group. However, compared with the 1.0% CoQ_10_ group, a significant reduction of 50.7% (*p* < 0.05) in TG levels was observed in the 1.0% BWA group, suggesting the higher efficacy of BWA in treating hypertriglyceridemia induced by HCD supplementation.

As shown in Figure 2C, the plasma HDL-C levels were the lowest in the HCD-only group (around 61 mg/dL) and were significantly increased by 1.6 (*p* < 0.001), 3.8 (*p* < 0.001) and 4.2 times (*p* < 0.001) in response to 0.1%, 0.5% and 1.0% BWA, respectively, indicating a substantial dose-dependent effect of BWA of increasing HCD-impaired HDL-C levels. Also, the 1.0% CoQ_10_ group displayed 1.8-fold (*p* < 0.001) higher HDL-C levels than the HCD control. Intriguingly, the effect exerted by 1.0% CoQ_10_ was nearly similar to the effect exerted by 0.1% BWA. Compared with the 1.0% BWA group, a significant 2.4-fold reduction (*p* < 0.01) in HDL-C levels was detected in the 1.0% CoQ_10_ group, suggesting a higher effectiveness of BWA in increasing HDL-C levels compared with CoQ_10_.

Concomitantly to HDL-C, the HCD-only group showed the lowest percentage of HDL-C in TC, around 10%, which was significantly increased by 19.2% (*p* < 0.05), 51.2% (*p* < 0.001) and 76.3% (*p* < 0.001) in response to 0.1%, 0.5% and 1.0% BWA, respectively (Figure 2D). Likewise, 1.0% CoQ_10_ consumption also significantly increased (*p* < 0.01, 25.8%) the HCD-induced decrease in HDL-C/TC (%), which was nearly similar to the HDL-C/TC (%) observed in 0.1% BWA. However, compared with 1.0% CoQ_10_, the 1.0% BWA group showed 2.9-fold higher (*p* < 0.01) HDL-C/TC (%) levels.

In contrast to HDL/TC (%), the HCD group exhibited a notably elevated TG/HDL-C ratio, which was significantly reduced (*p* < 0.001) by the consumption of BWA at concentrations ranging from 0.1% to 1.0% (Figure 2E). Similarly, 1.0% CoQ_10_ also considerably (*p* < 0.001) reduced the HCD-induced high levels of TG/HDL-C; however, compared with 1.0% BWA, 4.7-fold (*p* < 0.05) lower efficacy was noticed in the 1.0% CoQ_10_ group.

Likewise, BWA and CoQ_10_ exerted substantial effects (*p* < 0.001) of decreasing the HCD-induced increase in the LDL-C/HDL-C ratio; however, BWA, precisely at 1.0%, displayed significantly higher efficacy than 1.0% CoQ_10_ in balancing the HCD-affected LDL-C/HDL-C ratio (Figure 2F). These findings indicate that BWA co-supplementation provides significantly higher benefits than CoQ_10_ in stabilizing the lipid profile affected by HCD.

### 2.3. Change in Hepatic Damage Parameters

As shown in Figure 3, the HCD-only group showed the highest blood aspartate aminotransferase (AST) and alanine aminotransferase (ALT) levels, around 544.8 ± 3.4 IU/L and 365.7 ± 3.1 IU/L, respectively, suggesting that HCD consumption caused severe hepatic damage. Co-supplementation with BWA at 0.1%, 0.5% and 1.0% efficiently reduced the levels of AST by 14.4% (*p* < 0.05), 28.1% (*p* < 0.01) and 42.2% (*p* < 0.001), respectively, compared with the HCD-only group, testifying to the dose-dependent preventive impact of BWA against the HCD-induced increase in AST levels. Similarly, the blood ALT levels increased by the HCD decreased by 17.6% (*p* < 0.01), 26.4% (*p* < 0.001) and 45.3% (*p* < 0.001) in the 0.1%, 0.5% and 1.0% BWA groups, respectively.

The 1.0% CoQ_10_ group showed 23.5% (*p* < 0.01) and 15.9% (*p* < 0.01) lower AST and ALT levels than the HCD-only group. Compared with the 1.0% CoQ_10_ group, the 1.0% BWA group exhibited 24.5% (*p*< 0.05) and 34.9% (*p* < 0.01) reductions in AST and ALT levels, respectively. These results suggest that consumption of BWA, primarily at 1.0%, was remarkably more effective in ameliorating hepatic damage in hyperlipidemia than CoQ_10_.

### 2.4. Liver Size and Weight

The images of the whole liver collected from the zebrafish revealed a slightly enhanced liver size, indicating steatosis (fatty liver) in the HCD group. The HCD-induced large liver size was effectively averted with the consumption of BWA and CoQ_10_ (Figure 4). Consistent with these findings, liver size enlargement and increased weight were noticed in the HCD groups, which were significantly (*p* < 0.05) reduced following consumption of BWA and CoQ_10_. Interestingly, the 0.1% BWA group showed a similar effect to the 1.0% CoQ_10_ group in preventing liver size and weight enlargement.

### 2.5. Histological Analysis of Hepatic Inflammation

As shown in Figure 5A,E, H&E staining revealed that the HCD-only group showed the highest extent of neutrophil infiltration (indicated by the red arrows) and lipid droplet accumulation (indicated by the blue arrows). A marked accumulation of tiny lipid droplets (indicated by the green arrows) leading to vacuole formation in hepatocytes was prominently observed in the HCD-only group. Additionally, large lipid vesicles (indicated by the red arrows) caused nuclear shrinkage in certain regions, resulting in severe steatosis. In contrast, the 0.5% and 1.0% BWA groups exhibited significantly reduced infiltration of neutrophils and lipid droplet aggregation compared with the HCD group. However, the occasional presence of vacuolated hepatocytes and minor steatosis was noticed in the 0.5% and 1.0% BWA groups, though it was substantially less frequent compared with the HCD group. Furthermore, the quantification of the H&E-stained area revealed 57.8% (*p* < 0.05) and 79.2% (*p* < 0.05) smaller H&E-stained areas in the 0.5% and 1.0% BWA groups compared with the HCD-only group, suggesting a dose-dependent impact of BWA of preventing HCD-induced hepatic damage (Figure 5B,E). Interestingly, the 0.1% BWA group and the 1.0% CoQ_10_ group showed a non-significant effect (*p* > 0.05) on the inhibition of HCD-induced neutrophil infiltration and H&E-stained area, as shown in Figure 5A,B,E.

Immunohistochemistry (IHC) revealed that the HCD-only group showed the highest production of IL-6, as shown by the vital red conversion area (Figure 5C,D). At the same time, the 1.0% BWA group exhibited the lowest IL-6 levels, around 96.1% (*p* < 0.01) lower than the HCD-only group (Figure 5C,D,F). Co-consumption of 0.1% and 0.5% BWA caused significant decreases in IL-6 production of 75.1% (*p* < 0.05) and 86.6% (*p* < 0.01) compared with the HCD group, testifying to a dose-dependent effect of BWA against the inhibition of HCD-induced IL-6 production. Further, 1.0% CoQ_10_ consumption displayed a substantial IL-6 production inhibitory role, like the effect of the 0.1% BWA group. However, compared with 1.0% CoQ_10_ consumption, 6.8-fold (*p* < 0.05) higher efficacy was observed in response to 1.0% BWA consumption to curtail HCD-induced high IL-6 levels, testifying to the supremacy of BWA over CoQ_10_ in preventing hepatic inflammation measured in terms of IL-6 production.

### 2.6. Reactive Oxygen Species Generation and Cell Senescence in Liver

As shown in Figure 6A, the HCD-only group showed the most robust red (DHE) fluorescent intensity, depicting the utmost extent of hepatic reactive oxygen species (ROS) generation. On the contrary, the co-consumption of BWA effectively decreased ROS production, especially in the 1.0% BWA group, which exhibited around 80.2% (*p* < 0.001) less DHE fluorescent intensity than the HCD-only group (Figure 6A,C). On the other hand, the 0.1% BWA group and the 1.0% CoQ_10_ group showed a substantial preventive effect against HCD-induced ROS generation, manifested as significant reductions of 53.4% (*p* < 0.01) and 43.4% (*p* < 0.05) in DHE fluorescent intensity compared with the HCD-only group. On the other hand, compared with 1.0% CoQ_10_, the 1.0% BWA group showed a significant reduction of 62.1% (*p* < 0.05) in the DHE-stained area, attesting to the higher efficacy BWA in restricting HCD-induced ROS generation.

As shown in Figure 6B, senescence-associated β-galactosidase (SA-β-gal) staining revealed that the HCD-only group exhibited the most severe cell senescence, around 54.5% of SA-β-gal-positive cells, suggesting that the 20 weeks of consumption of a high-cholesterol diet resulted in severe cellular senescence in the liver. The consumption of BWA and CoQ_10_ substantially prevented HCD-induced senescence (Figure 6B,D). Compared with 1.0% CoQ_10_, 1.0% BWA exhibited a significant reduction of 84.2% (*p* < 0.001) in SA-β-gal-positive cells, demonstrating BWA enhanced efficacy over CoQ_10_ in mitigating HCD-induced cellular senescence.

### 2.7. Histological Analysis of Kidneys

As depicted in Figure 7A, the HCD group showed loosely arranged and sparsely populated proximal and distal tubular structures with frequent lumen debris (as indicated by the red arrows). On the contrary, the consumption of BWA, mainly 1.0% BWA, restored kidney histology impaired by the HCD. Likewise, 0.1% and 0.5% BWA also displayed a preventive effect against HCD-induced kidney impairment; however, the rare presence of cellular debris (as indicated by the red arrows) was observed in the tubular cast. Further, 1.0% CoQ_10_ consumption displayed a substantial preventive effect against HCD-induced kidney damage, nearly similar to that observed in the 0.5% BWA group.

DHE staining revealed massive ROS production in the HCD group, which was prevented by the consumption of 0.5% BWA, 1.0% BWA and 1.0% CoQ_10_, evidenced by 45.6%, 57.7% and 45.1% lower DHE fluorescent intensity compared with the HCD group (Figure 7B,D). However, the difference concerning the HCD-only group was statistically non-significant.

SA-β-gal staining suggested higher occurrence of senescent cells in the HCD group. The consumption of BWA and CoQ_10_ efficiently inhibited HCD-induced cellular senescence (Figure 7C,E). The SA-β-gal-stained area obtained from the 0.5% and 1.0% BWA groups revealed 87.1% (*p* < 0.001) and 88.3% (*p* < 0.001) lower levels of senescent cells compared with the HCD group. Also, a substantial effect, with a 67.3% (*p* < 0.01) smaller SA-β-gal-stained area, was noticed in the 1.0% CoQ_10_ group compared with the HCD group. Compared with COQ_10_, a 64.5% smaller SA-β-gal-stained area was marked in the 1.0% BWA group, underscoring the superiority of BWA over the CoQ_10_ in mitigating HCD-induced senescence in the kidneys.

### 2.8. Testicular Cell Analysis

The H&E staining of the testicular sections (Figure 8A,B) showed that the HCD group had loosely arranged seminiferous tubules, occasional disruption of lamina basal membranes (indicated by the red arrows) and increased interstitial tissue area (around 33.8%) between the seminiferous tubules. Additionally, irregularly distributed spermatogonia, spermatocytes and spermatozoa were frequently observed, including void space in the lumen.

The co-consumption of BWA improved the morphological features of testicular cells, as reflected by intact seminiferous tubules with a decrease in interstitial space between the seminiferous tubules, especially in the 1.0% BWA group, which showed a noticeable 36.4% (*p* < 0.001) decrease in interstitial space area compared with the HCD group (Figure 8A,B,E). Additionally, spermatocytes, spermatozoa and spermatogonia were arranged appropriately, with the least void space in the lumen of seminiferous tubules. Notably, higher prevalence of spermatozoa (mature sperm) was noticed in the 1.0% BWA group than the HCD group.

Like the BWA effect, CoQ_10_ consumption also showed a substantial improvement in testicular cell morphology, as evidenced by a reduction of around 16.3% (*p* < 0.05) in interstitial space compared with the HCD group. However, 1.0% BWA exerted a substantially higher effect than 1.0% CoQ_10_ in terms of preventing HCD-induced testicular damage, as indicated by 24.1% (*p* < 0.01) lower interstitial space in the seminiferous tubules, indicating comparative higher efficacy of BWA compared with CoQ_10_.

The DHE staining outcomes reveal massive ROS production in the HCD group, which was significantly (*p* < 0.001) prevented by the consumption of BWA and CoQ_10_ (Figure 8C,F). Compared with CoQ_10_, BWA consumption at 0.1%, 0.5% and 1.0% displayed 23.7%, 49.4% and 63.3% lower DHE fluorescent intensity, suggesting the functional superiority of BWA over CoQ_10_ in maintaining HCD-induced high oxidative stress.

SA-β-gal staining revealed that the HCD-only group showed the highest blue color intensity and blue color-stained area corresponding to SA-β-gal-positive senescent cells, which was substantially prevented by the consumption of BWA in a dose-dependent manner (Figure 8D,G). As shown in Figure 8D,G, the HCD-only group showed an SA-β-gal-stained area of 14.1%, which was reduced to 4.0%, 1.4% and 0.2% by the consumption of 0.1% BWA, 0.5% BWA and 1.0% BWA, respectively. Likewise, CoQ_10_ also significantly prevented HCD-induced cellular senescence. However, compared with the BWA, the efficacy of CoQ_10_ in preventing HCD-induced cellular senescence was much lower. As depicted in Figure 8D,G, a 49-fold (*p* < 0.001) smaller senescent area was detected in the 1.0% BWA group compared with the 1.0% CoQ_10_ group, attesting to the higher potency of BWA in mitigating HCD-induced testis senescence.

### 2.9. Ovarian Cell Analysis

As shown in Figure 9A,D, the HCD-only group showed the highest previtellogenic oocyte content (~93%) and the lowest mature-vitellogenic oocyte content (~2%), indicating the adverse effect of HCD supplementation on the impairment of oocyte development. The co-consumption of BWA improved HCD-impaired oocyte development in a dose-dependent manner, especially at 1.0%. The BWA group showed the lowest previtellogenic oocyte content (~73%) and the highest mature-vitellogenic oocyte content (~10%). Also, CoQ_10_ consumption substantially affected the restoration of HCD-altered oocyte counts. Between the 1.0% consumption of BWA and CoQ_10_, BWA displayed a significant 12.1% (*p* < 0.05) reduction in previtellogenic oocyte content, indicating the superior functionality of BWA in addressing HCD-induced ovary impairment.

The findings of DHE staining show the influential role of BWA in overcoming HCD-induced ROS production (Figure 9B,E). Significant reductions of 31.2%, 43.4% and 51.5% in DHE-stained area in response to 0.1%, 0.5% and 1.0% BWA compared with the HCD-only group testify to the dose-dependent effect of BWA of diminishing HCD-induced ROS production. Also, CoQ_10_ consumption displayed a substantial preventive effect on HCD-induced ROS production; however, compared with 1.0% BWA, the effect exerted by 1.0% CoQ_10_ was significantly 41.1% lower (*p* < 0.001), indicating comparatively lower efficacy of CoQ_10_ with respect to BWA in reducing ROS production.

SA-β-gal staining suggested higher prevalence of cellular senescence in the ovarian section of the HCD group, which was efficiently mitigated by the co-consumption of both BWA and CoQ_10_ (Figure 9C,F). However, compared with BWA, the efficacy of CoQ_10_ was lower in preventing HCD-induced senescence in the ovaries. Compared with 1.0% CoQ_10_, significant reductions of 80.1% (*p* < 0.001) and 61.5% (*p* < 0.001) in SA-β-gal-stained area were found in the 1.0% BWA group, attesting to its importance in mitigating HCD-induced ovary impairment.

## 3. Discussion

In the present study, the comparative effects of BWA and CoQ_10_ consumption were examined on the blood lipid profile, body weight, survivability and function of vital organs of hyperlipidemic zebrafish. Zebrafish was selected as a model organism due to its high genome similarity to humans [22]. Also, the key events and essential receptors and enzymes of lipid metabolism in zebrafish are very similar to those in human beings, thus making it an excellent model organism for assessing the impact of drugs and nutraceuticals on dyslipidemia [23].

The established role of HCD supplementation in dyslipidemia in zebrafish [7,24,25] and other model organisms has been well elucidated [26,27]. Consistently, we have also noticed the impact of the HCD on the increase in TC, TG and LDL-C and decrease in HDL-C levels in zebrafish. Supplementation with both BWA and CoQ_10_ was found to be adequate to counter HCD-induced dyslipidemia by effectively managing the altered TC, TG, LDL-C and HDL-C levels. Nevertheless, the comparative analysis suggested a much inferior effect of CoQ_10_ compared with BWA in managing the disrupted plasma lipid profile caused by the HCD. A noteworthy 2.3-fold increase in HDL-C levels was observed in response to the consumption of 1.0% BWA compared with that observed in CoQ_10_. The outcomes enforced the functional superiority of BWA over CoQ_10_ in the management of HCD-induced dyslipidemia. The results are aligned with the accumulating literature depicting the impact of CoQ_10_ on decreasing TC, TG and LDL-C levels and increasing HDL-C levels [28]. However, the literature has revealed a variation in the countering of the dyslipidemic effect of CoQ_10_ in response to the used dose of CoQ_10_. Moreover, studies have documented the non-effectiveness of CoQ_10_ at the tested concentration of 150 mg/day in diabetic patients [19]. Contrary to the available literature for CoQ_10_, a limited study describes the effect of BWA on the plasma lipid profile. In that study, BWA displayed a substantial impact on the increase in HDL-C levels hampered by exposure to ethanol [7], supporting the present finding concerning BWA’s positive effect on the increase in HDL-C levels. The presence of LCAAs, such as hexacosanol, triacontanol and octacosanol, in BWA is the key contributor responsible for the beneficial effect of BWA on the management of the HCD-affected plasma lipid profile. This notion is supported by the previous finding depicting the inhibitory role of triacontanol [29] and hexacosanol [30] in cholesterol biosynthesis. Research has been documented the visible effect of hexacosanol on the activation of AMP-activated protein kinase (AMPK), which induced the inactivation of 3-hydroxy-3-methyl-glutaryl-coenzyme A reductase (HMG-CoA), a principal rate-limiting enzyme for cholesterol biosynthesis [30]. Additionally, hexacosanol has a negative impact on the nuclear translocation of sterol regulatory element-binding protein-2 (SREBP-2), which regulates the production of HMG-CoA at the transcriptional level, thus inhibiting cholesterol biosynthesis [30]. A similar effect of octacosanol (a major constituent of BWA) of inhibiting cholesterol biosynthesis has been observed [31,32].

Beyond improving the plasma lipid profile, BWA demonstrated a notable ability to protect the liver from damage associated with the HCD. Substantially reduced neutrophil infiltration, prevention of fatty liver changes and hepatic IL-6 generation were observed in the BWA group in a dose-dependent manner. Specifically, 1.0% BWA displayed a substantial hepatoprotective role, significantly higher than the effect exerted by the consumption of 1.0% CoQ_10_. Also, compared with CoQ_10_, BWA displayed higher efficacy in preventing HCD-induced oxidative stress and cellular senescence in the liver. The findings are consistent with previous studies that have demonstrated the hepatoprotective effect of BWA [11,33]. In one such study, BWA consumption for 6 months substantially improved fatty liver changes and insulin resistance in patients with non-alcoholic fatty liver disease (NAFLD) [33].

BWA’s antioxidant and anti-inflammatory effect, demonstrated in several preclinical and clinical studies, ascribe a gastro-protective role to this substance [10,34,35,36]. The antioxidant role of BWA mediated by the scavenging of hydroxyl radicals and by modulating cellular antioxidant catalase, superoxide dismutase (SOD) and glutathione peroxidase (GSH-PX) has been found to be a key event in protecting the gastric mucosa from a variety of stimuli [9,35]. The better hepatoprotective activity of BWA compared with CoQ_10_ can be justified by earlier findings documenting a higher antioxidant impact of BWA compared with CoQ_10_ on the scavenging of free radicals and the inhibition of oxidative damage of lipoproteins [6]. Moreover, compared with CoQ_10_, BWA efficiently inhibits CML-induced oxidative stress and protects zebrafish embryos from apoptotic cell death [6], indicative of BWA’s better cellular antioxidant role compared with CoQ_10_. It has also been described that BWA substantially enhances the activity of the HDL-associated antioxidant paraoxonase (PON)-1, while CoQ_10_ failed to display any such effect [6]. This might be an additional reason behind the better antioxidant role of BWA compared with CoQ_10_ in mitigating HCD-induced challenges and oxidative stress. Furthermore, BWA consumption (100 mg BWA/day for 12 weeks) showed substantial potential to reduce malondialdehyde (MDA) and total hydroperoxide levels and augmenting total antioxidant status in humans, strengthening the remarkable antioxidant impact of BWA [6].

Increased liver size is commonly linked to fatty liver disease [37], and in our study, we observed a similar increase in liver size following HCD consumption. However, BWA treatment notably reduced this increase, indicative of its hepatoprotective effect, as further supported by hepatic histological results. The hepatoprotective nature of BWA, as observed from the histological analysis, is also confirmed by the diminished plasma AST and ALT levels, which are important hepatic function biomarkers [38,39,40].

The lower prevalence of AST and ALT in the 1.0% BWA group compared with CoQ_10_ suggests a greater hepatoprotective function of BWA, consistent with the findings of the hepatic histological outcomes. We believe that the reduced AST and ALT levels could be correlated with elevated HDL-C and low TC levels in response to BWA. This notion is in accordance with earlier findings that establish a negative correlation between elevated HDL-C [40] and low TC levels [41] and AST and ALT levels, strengthening the current findings that suggest that BWA, compared with CoQ_10_, has substantial effects on the increase in HDL-C, decrease in TG levels and consequently lower AST and ALT levels, indicating better hepatic health.

BWA prevents the fatty liver changes induced by an HCD; this may be attributed to the presence of hexacosanol, which has been identified to stimulate autophagy commenced by the upregulation of an autophagy-related gene (ATG16L) and anti-microtubule-associated protein 1A/1B light chain 3 (LC3-II) [30], resulting in the inhibition of hepatic lipid accumulation and subsequent fatty liver changes [30,42]. Elevated levels of IL-6 in response to HCD consumption may be associated with excessive ROS generation, as several reports depict the induction of the inflammatory pathway induced by oxidative stress [43]. Also, the induction of IL-6 might be a major contributor that aggravates HCD-induced dyslipidemia. The notion is supported by the findings disclosing elevated TG levels in response to IL-6 production [44]. Further, many inflammatory disorders, such as rheumatoid arthritis (RA), systemic lupus erythematosus (SLE) and psoriasis, substantially affect the blood lipid profile, indicating the tight association between inflammation and an altered lipid profile [45,46,47]. As BWA showed higher effectiveness than CoQ_10_ in restricting HCD-induced high IL-6 production, it might be a key component of the mechanism behind a high degree of hepatic protection and the management of the plasma lipid profile. Our results are in line with earlier studies describing BWA’s potential as an anti-inflammatory agent that prevents inflammatory disorders (arthritis) in mice [48] and humans [49].

The association of ROS/oxidative stress with cellular senescence has already been described [50,51]. BWA, owing to its better inhibitory impact than CoQ_10_ on cellular ROS generation, exerts an ameliorating effect on oxidative stress and leads to subsequent prevention of cellular senescence.

HCD-induced adverse effects have been described as damage to the kidneys [52] and reproductive system [20,53]. The adverse impact of HCDs on the testicular structures, precisely on the seminiferous tubules and the interstitial area, has been documented [54], and we have also observed a similar detrimental effect in zebrafish. Noticeable superior kidney, testis and ovary protective effects exerted by BWA compared with CoQ_10_ by curtailing HCD-induced ROS generation and cellular senescence were observed in zebrafish during the 20 weeks of the consumption study. We believe that kidney protection against HCD-induced damage is attributed to BWA’s higher antioxidant efficacy, which maintains cellular oxidative stress, which has been described as a key event in preventing cellular senescence [55] in a variety of organs. Additionally, the critical role of hexacosanol (a component of BWA) has been illustrated as the modulation of nitric oxide synthetase and protein kinase C (PKC), leading to the prevention of glomerular sclerosis and improvement in diabetic nephropathy [56,57]. To the best of our knowledge, this is the first study describing the protective effect of BWA against cholesterol-induced impairment of the liver, kidney and reproductive organs.

## 4. Materials and Methods

### 4.1. Materials

A typical blend of six long-chain aliphatic alcohols (LCAAs; C24-C34) containing beeswax alcohol (BWA; Raydel^®^) was extracted from a block of wax of honeybees (mainly mellifera linage), which was complementary provided by Raydel**^®^** Pty, Ltd., Thornleigh, NSW, Australia. The BWA (Batch#: 330020123) was >86% pure and contained 6.1% tetracosanol, 10.7% hexacosanol, 13.8% octacosanol, 30.5% triacontanol, 22.1% dotriacontanol and 2.9% tetratriacontanol. A certificate of chemical analysis is provided in Appendix A. All the other chemicals and reagents used in the present study were of analytical grade and used as supplied unless otherwise stated. Appendix A contain a detailed list of the chemicals and reagents used.

### 4.2. Zebrafish Maintenance

The adult (~16 weeks aged) wild zebrafish (*Danio rerio*) AB strain of mixed gender (male and female) was procured from a local market and fed normal tetrabit (ND; Gmbh D49304, Melle, Germany). The ND was composed of 47.5% crude protein, 6.5% crude fat, 2.0% crude fiber, 10.5% crude ash, vitamin A (29.77 IU/g), vitamin D3 (1.86 IU/g), vitamin E (0.2 mg/g) and vitamin C (0.13 mg/g). The zebrafish were housed in an aerated water tank maintained at 28 °C with constant water supply and a photoperiod of 10 h of darkness and 14 h of light. The zebrafish were maintained following the prescribed rules of care and use of laboratory animals [58], and the study protocols were approved by the Committee of Animal Care and Use of Raydel Research Institute (approval code RRI-23-001; approval date 27 July 2023).

### 4.3. Preparation of Different Diets

The different BWA or CoQ_10_ formulated diets were prepared by incorporating BWA or CoQ_10_ in a high-cholesterol diet (HCD). To prepare the HCD, normal tetrabit (ND) was mixed with cholesterol (final concentration of 4%, *w*/*w*) [59]. In brief, the ND (800 g) was supplemented with 32 g of cholesterol (equivalent to 4% *w*/*w*) and mixed (by spatula) to distribute the cholesterol. Subsequently, an organic solvent (chloroform) was added to the mixture (ND + cholesterol) and agitated mechanically to properly distribute the cholesterol. The chloroform was completely evaporated in the fume hood to obtain ND supplemented with cholesterol, which was named the HCD. The HCD (25 mg) from three different locations was collected to ascertain nearly similar cholesterol distribution. The cholesterol was extracted from the samples by vigorous agitation with chloroform. The extracted cholesterol was quantified by using a commercial cholesterol diagnostic kit (cholesterol, T-CHO, Clean TS-S; Walko Pure Chemicals, Osaka, Japan).

The HCD was individually supplemented with BWA (final concentration of 0.1%, 0.5% and 1.0%, *w*/*w*) to form three different BWA-enriched diets named HCD + 0.1% BWA, HCD + 0.5% BWA and HCD + 1.0% BWA, respectively. Similarly, CoQ_10_ (final concentration of 1.0%, *w*/*w*) was mixed with the HCD to prepare a CoQ_10_-enriched diet named HCD + 1.0% CoQ_10_. Detailed proportions of the tetrabit, cholesterol, BWA and CoQ_10_ used to prepare different diets are included in Table 2. The selection of the used diets (0.1–1.0%, *w*/*w*) was based on preliminary studies, where we tested the effect of different amounts of BWA and found that 1.0% BWA as a maximum diet addition exerted beneficial effects on hyperlipidemia zebrafish [60].

### 4.4. Consumption of Different Diets by Zebrafish

All the zebrafish (~16 weeks old) were exclusively fed the HCD for 4 weeks before segregation into different groups. After four weeks of feeding on the HCD, the zebrafish (n = 280) were arbitrarily allocated to five distinct groups (n = 56/group) (Figure 10). The zebrafish in group I were continuously fed only the HCD. The zebrafish in groups II, III and IV were fed HCD + 0.1% BWA, HCD + 0.5% BWA and HCD + 1.0% BWA, respectively. The group V zebrafish were fed HCD + 1.0% CoQ_10_. Each group’s zebrafish (n = 56) were divided into four sets (n = 14/tank) and fed the specified diets for 20 weeks. Each set (n = 14) of different groups received 140 mg of the respective diets/tank twice a day in the morning and evening (i.e., a total of 280 mg/day, ~equivalent to 20 mg/day/zebrafish). Over the 20-week consumption period, zebrafish survivability was monitored every day.

The body weight of zebrafish in different groups was measured at 0, 4, 8, 12, 16 and 20 weeks by using the gravimetric method. Body weight was analyzed after anesthetizing the zebrafish by submerging them into 0.1% 2-phenoxyethanol. The surface of the anathematized zebrafish was dried by using filter paper, and the weight was measured with an electronic weighing balance (Ohaus, Parsippany-Troy Hills, NJ, USA); immediately, the zebrafish were transferred into their tank. For the body weight analysis, one set of zebrafish (n = 14) from each group were used. Notably, at different time points (0–20 weeks), similar sets of the respective groups were used to analyze body weight.

### 4.5. Collection of Blood and Organs

After 20 weeks of consumption of different diets, the zebrafish were sacrificed by using hypothermic shock [59], and blood was immediately collected in EDTA-prewashed tubes and subjected to centrifugation to obtain plasma. The plasma obtained from the different groups was analyzed by utilizing commercial kits to assess total cholesterol, triglyceride, HDL and the hepatic function enzymatic markers AST and ALT. Appendix A provides the detailed blood analysis procedure. The organs (liver, kidney, testes and ovaries) of the sacrificed zebrafish were surgically removed under an optical microscope (at 10× magnification) and preserved separately in 10% formalin.

### 4.6. Histology and Immunohistochemistry (IHC)

The tissue sections (liver, kidney, testes and ovaries) were embedded in FSC22 clear frozen compound (Leica) and subsequently processed for cryo-tissue sectioning by using a Leica CM 1510S cryo-microtome (Leica Biosystem, Nussloch, Germany). A thin section (7 μm) was sliced and processed for hematoxylin and eosin (H&E) staining [61]. The histological changes in the different H&E-stained tissue sections were analyzed under a microscope (Motic microscopy PA53MET; Hong Kong, China) equipped with a digital camera to examine neutrophil infiltration, vacuolation due to small lipid droplets accumulation in the hepatocytes, hepatic ballooning and steatosis (fatty liver changes). The H&E images of the hepatic tissue captured were processed by using Image J software (http://rsb.info.nih.gov/ij; accessed on 16 January 2023; version 1.53) to quantify the number of neutrophils (stained dark violet to blue) and the H&E-stained area.

For the testis sections, the H&E-stained area was visualized to examine spermatogenesis, seminiferous tubules and interstitial space between the seminiferous tubules [62,63]. The interstitial space in the H&E-stained images was quantified by using Image J software after converting the images into RGB stack according to a threshold level [of lower limit (220) and upper limit (255)]. A minimum of three images from each group were examined for the histological analysis of testicular tissue.

For the ovary sections, the previtellogenic, early-vitellogenic and mature-vitellogenic stages, as described previously [64,65], were analyzed. The classification for the previtellogenic stage is follicles measuring 250 μm in diameter or smaller, including the smallest previtellogenic follicles and those in the peri-nucleolar stage. Follicles of 250–500 μm in diameter were classified as being in the early-vitellogenic stage. Follicles of 500 μm in diameter or larger were assigned to the mature-vitellogenic stage, including follicles characterized by yolk-filled alveoli distributed throughout the ooplasm and vitellogenic follicles.

Cellular senescence in the tissue sections was examined by the senescence-associated β-galactosidase (SA-β-gal) assay [66]. In brief, the tissue section (7 μm thick) was covered with 5-bromo-4-chloro-3-indolyl-β-D-galactopyranoside (X-gal) overnight (16 h) and visualized microscopically for the detection of blue-stained senescent-positive cells.

Interleukin (IL)-6 in the hepatic section was detected by IHC staining employing an anti-IL-6 primary antibody and enzyme-linked anti-IL-6 antibodies following an earlier described method [67]. IL-6 quantification was performed by employing image J software by converting IL-6-stained images into RGB stack according to threshold levels [of lower limit (20) and upper limit (120)] to minimize the inclusion of background staining. All the images were processed at the same threshold values. The detailed procedure for IHC staining is provided in Appendix A.

### 4.7. Dihydroethidium (DHE) Fluorescent Staining

The tissue sections (7 μm thick) were stained for 30 min with DHE (final concentration of 30 μM) [68]. Subsequently, the stained sections were washed with phosphate-buffered saline (PBS) and visualized under a fluorescent microscope at the excitation and emission wavelengths of 585 nm and 615 nm.

### 4.8. Statistical Analysis

One-way analysis of variance (ANOVA) followed by Tukey’s post hoc analysis, performed by employing SPSS software (version 29.0; Chicago, IL, USA), was used to establish the statistical difference between the groups. The experimental values for each experiment were obtained from three replicate experiments and presented as the average values ± standard errors of the mean (SEM).

## 5. Conclusions

This study compared the functional difference between BWA and CoQ_10_ against the detrimental effects caused by the consumption of an HCD. After 20 weeks of consumption, BWA (final concentration of 1.0%, *w*/*w*) showed a significantly better effect than CoQ_10_ (final concentration of 1.0%, *w*/*w*) in preventing HCD-induced high TC and TG levels. Consistently, BWA increased HDL-C levels more than CoQ_10_. A substantially better effect compared with CoQ_10_ in preventing HCD-induced liver, kidney and reproductive organ damage was observed in response to BWA consumption. Also, higher efficacy of BWA compared with CoQ_10_ was noticed in countering HCD-induced ROS generation, which lead to the protection of liver, kidney and reproductive organs. Compared with CoQ_10_, BWA displayed substantially higher anti-inflammatory activity by curtailing HCD-induced hepatic IL-6 production and preventing fatty liver changes. This study confirms the functional superiority of BWA over CoQ_10_; the former can be used as a nutraceutical to prevent dyslipidemia and associated adverse disorders.

## Figures and Tables

**Figure 1 pharmaceuticals-17-01434-f001:**
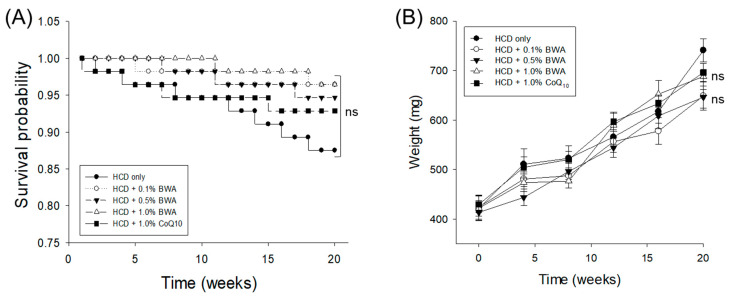
Survival probability (**A**) and body weight change (**B**) in zebrafish across the different groups during 20 weeks of consumption of the respective diets. HCD stands for the high-cholesterol diet, while BWA and CoQ_10_ stand for beeswax alcohol and coenzyme Q_10_, respectively. “ns” represents the non-significant difference between the groups. The Kaplan–Meier survival analysis followed by the log-rank test was used to determine the survivable probability curve, while ANOVA followed by Tukey’s post hoc analysis was used for the statistical analysis of the body weight.

**Figure 2 pharmaceuticals-17-01434-f002:**
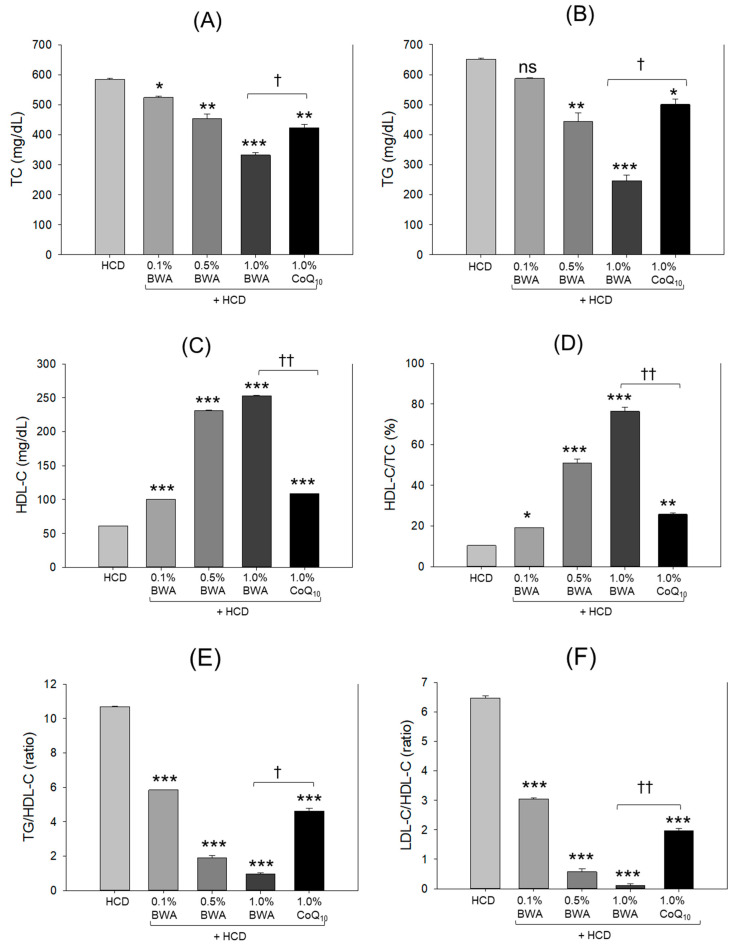
Quantification of (**A**) total cholesterol, (**B**) triglyceride, (**C**) HDL-C, (**D**) HDL-C/TC (%), (**E**) TG/HDL-C and (**F**) LDL-C/HDL-C among the different groups after 20 weeks of consumption of the respective diets. TC, TG, HDL-C, LDL-C, HCD, BWA and CoQ_10_ stand for total cholesterol, triglyceride, high-density-lipoprotein cholesterol, low-density-lipoprotein cholesterol, high-cholesterol diet, beeswax alcohol and coenzyme Q_10_, respectively. * (*p* < 0.05), ** (*p* < 0.01) and *** (*p* < 0.001) depict the level of significance with respect to the HCD group, while ^†^ (*p* < 0.05) and ^††^ (*p* < 0.01) highlight the significance between the 1.0% BWA and 1.0% CoQ_10_ groups.

**Figure 3 pharmaceuticals-17-01434-f003:**
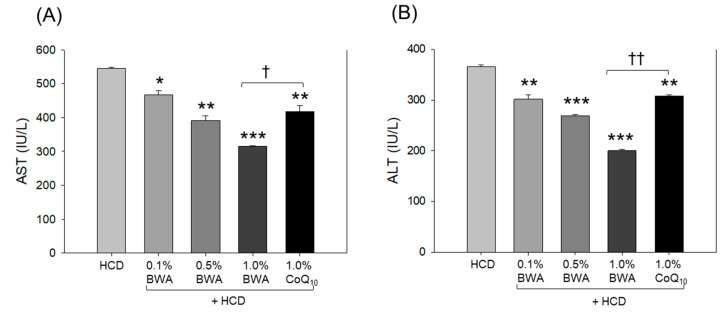
Quantification of the levels of the hepatic function biomarkers (**A**) aspartate aminotransferase and (**B**) alanine aminotransferase in the plasma of zebrafish after 20 weeks of consumption of different diets. HCD stands for the high-cholesterol diet, while BWA and CoQ_10_ stand for beeswax alcohol and coenzyme Q_10_, respectively. * (*p* < 0.05), ** (*p* < 0.01) and *** (*p* < 0.001) depict the level of significance with respect to the HCD group, while ^†^ (*p* < 0.05) and ^††^ (*p* < 0.01) highlight the significance between the 1.0% BWA and 1.0% CoQ_10_ groups.

**Figure 4 pharmaceuticals-17-01434-f004:**
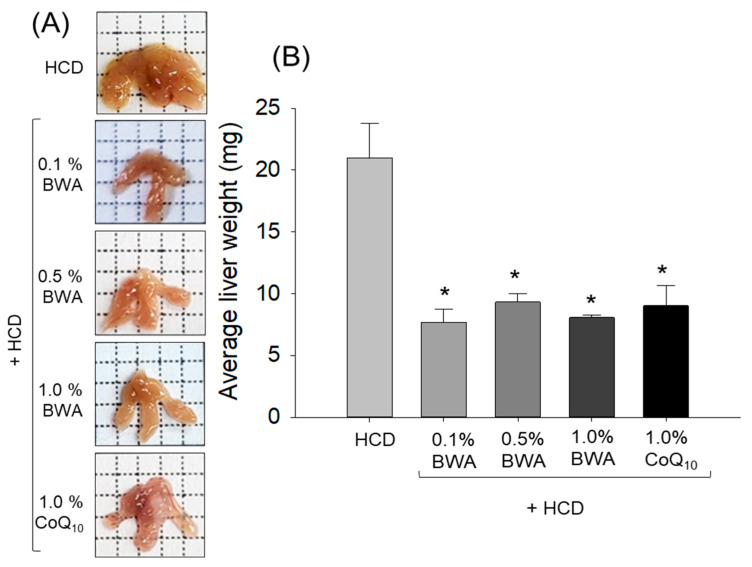
Comparison of liver morphology and size among zebrafish after 20 weeks of consumption of different diets. (**A**) Images of the whole liver and (**B**) liver weight of zebrafish following 20 weeks of consumption of different diets. HCD stands for the high-cholesterol diet, while BWA and CoQ_10_ stand for beeswax alcohol and coenzyme Q_10_, respectively. * (*p* < 0.05) depicts the level of significance with respect to the HCD group.

**Figure 5 pharmaceuticals-17-01434-f005:**
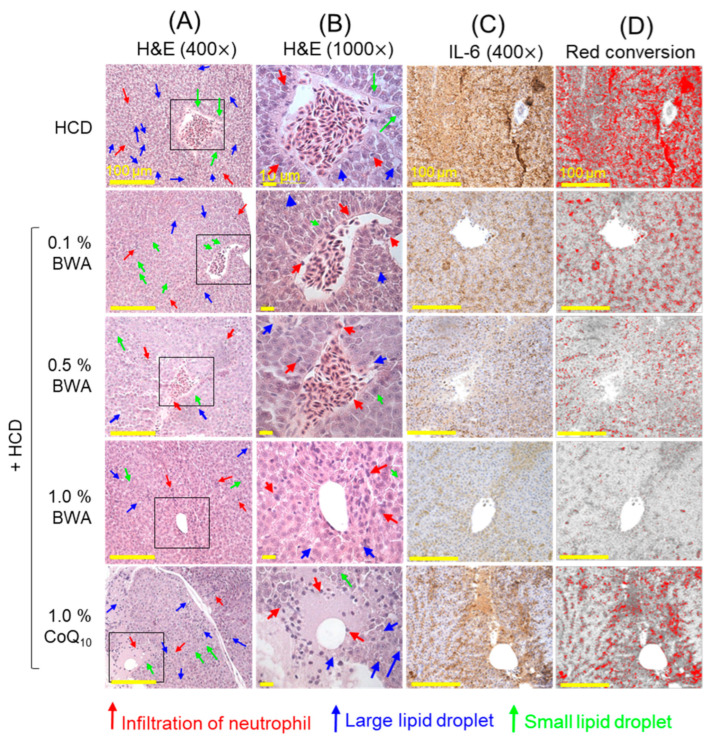
Hepatic histology of zebrafish after 20 weeks of consumption of different diets. (**A**,**B**) Hematoxylin and eosin (H&E) staining at 400× and 1000× magnifications; red and blue arrows indicate neutrophils and lipid droplets. (**C**) Interleukin (IL)-6-stained area examined by immunohistochemistry (IHC) and (**D**) visibility of IL-6-stained (brown color) area enhanced by its conversion into the red color (at brown color threshold of 20–120) performed by employing Image J software, version 1.53 [scale bar, 100 μm]. Quantification of (**E**) H&E-stained area and neutrophils counts and (**F**) IL-6-stained area. HCD stands for the high-cholesterol diet, while BWA and CoQ_10_ stand for beeswax alcohol and coenzyme Q_10_, respectively. * (*p* < 0.05), ** (*p* < 0.05) and *** (*p* < 0.001) depict the level of significance for the H&E- and IL-6-stained areas, while ^#^ (*p* < 0.05) depicts the significance level for neutrophil counts with reference to the HCD group. ^†^ (*p* < 0.05) and ^††^ (*p* < 0.01) depict the statistical difference between the 1.0% BWA and 1.0% CoQ_10_ groups; the non-significant differences between the groups are represented by “ns”.

**Figure 6 pharmaceuticals-17-01434-f006:**
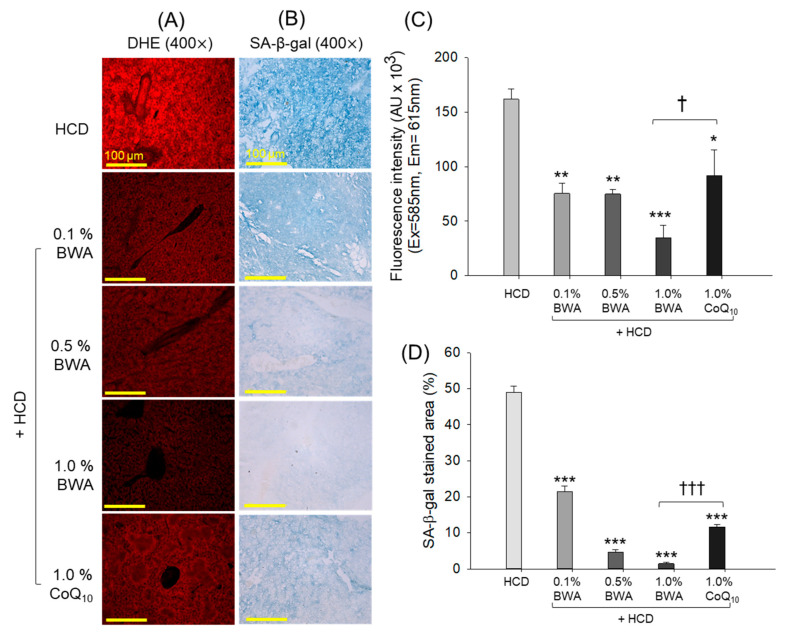
Detection of reactive oxygen species (ROS) generation and cellular senescence in the hepatic tissue of zebrafish following 20 weeks of consumption of different diets. (**A**) Dihydroethidium (DHE) fluorescent staining was used to detect ROS. (**B**) Senescence-associated β-galactosidase (SA-β-gal) staining [scale bar, 100 μm]. (**C**) Image J software-derived DHE fluorescent intensity quantification. (**D**) SA-β-gal-stained area quantification. HCD stands for the high-cholesterol diet, while BWA and CoQ_10_ stand for beeswax alcohol and coenzyme Q_10_, respectively. * (*p* < 0.05), ** (*p* < 0.01) and *** (*p* < 0.001) depict the level of significance with reference to the HCD group. ^†^ (*p* < 0.05) and ^†††^ (*p* < 0.001) highlight the significance with respect to the 1.0% BWA group.

**Figure 7 pharmaceuticals-17-01434-f007:**
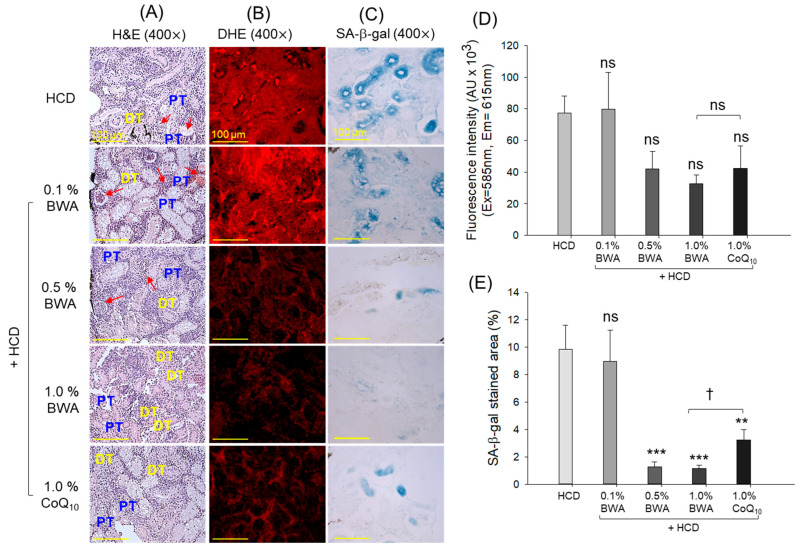
Analysis of kidney histology of zebrafish at 20 weeks of consumption of different diets. (**A**) Hematoxylin and eosin staining: proximal and distal tubules are represented by PT and DT, respectively, and the red arrows indicate luminal debris. (**B**) Dihydroethidium (DHE) fluorescent staining was used to detect reactive oxygen species (ROS). (**C**) Senescence-associated β-galactosidase (SA-β-gal) staining [scale bar, 100 μm]. (**D**) Image J software-derived DHE fluorescent intensity quantification. (**E**) Quantification of SA-β-gal-stained area. HCD stands for the high-cholesterol diet, while BWA and CoQ_10_ stand for beeswax alcohol and coenzyme Q_10_, respectively. ** (*p* < 0.01) and *** (*p* < 0.001) depict the level of significance with reference to the HCD group. ^†^ (*p* < 0.05) highlights the significance between the 1.0% BWA and 1.0% CoQ_10_ groups; the non-significant differences between the groups are represented by “ns”.

**Figure 8 pharmaceuticals-17-01434-f008:**
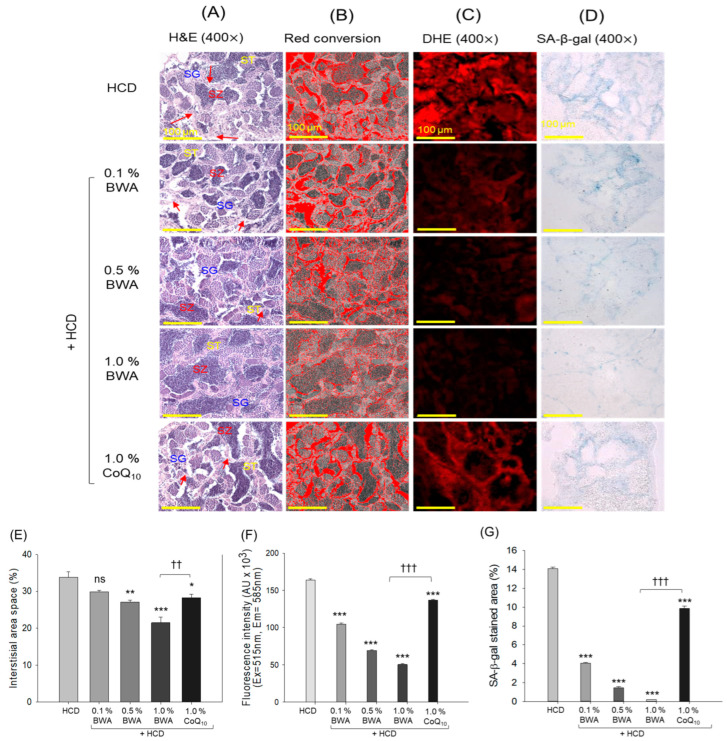
Histological examination of zebrafish testes following 20 weeks of consumption of different diets. (**A**) Hematoxylin and eosin (H&E) staining: spermatocytes, spermatozoa and spermatogonia are represented by ST, SZ and SG, respectively; disruption of lamina basal membranes is indicated by the red arrows. (**B**) The interstitial area was red-converted into the H&E section by using Image J software (at white color threshold of 220–255) to enhance visibility. (**C**) Dihydroethidium (DHE) fluorescent staining used to detect reactive oxygen species (ROS) and (**D**) senescence-associated-β-galactosidase (SA-β-gal) staining [scale bar, 100 μm]. (**E**) Quantification of interstitial area performed by employing Image J software. (**F**,**G**) Quantification of DHE fluorescent intensity and SA-β-gal-stained area, respectively. HCD stands for the high-cholesterol diet, while BWA and CoQ_10_ stand for beeswax alcohol and coenzyme Q_10_, respectively. * (*p* < 0.05), ** (*p* < 0.01) and *** (*p* < 0.001) depict the level of significance with reference to the HCD group. ^††^ (*p* < 0.01) and ^†††^ (*p* < 0.001) highlight the significance between the 1.0% BWA and 1.0% CoQ_10_ groups; the non-significant differences between the groups are represented by “ns”.

**Figure 9 pharmaceuticals-17-01434-f009:**
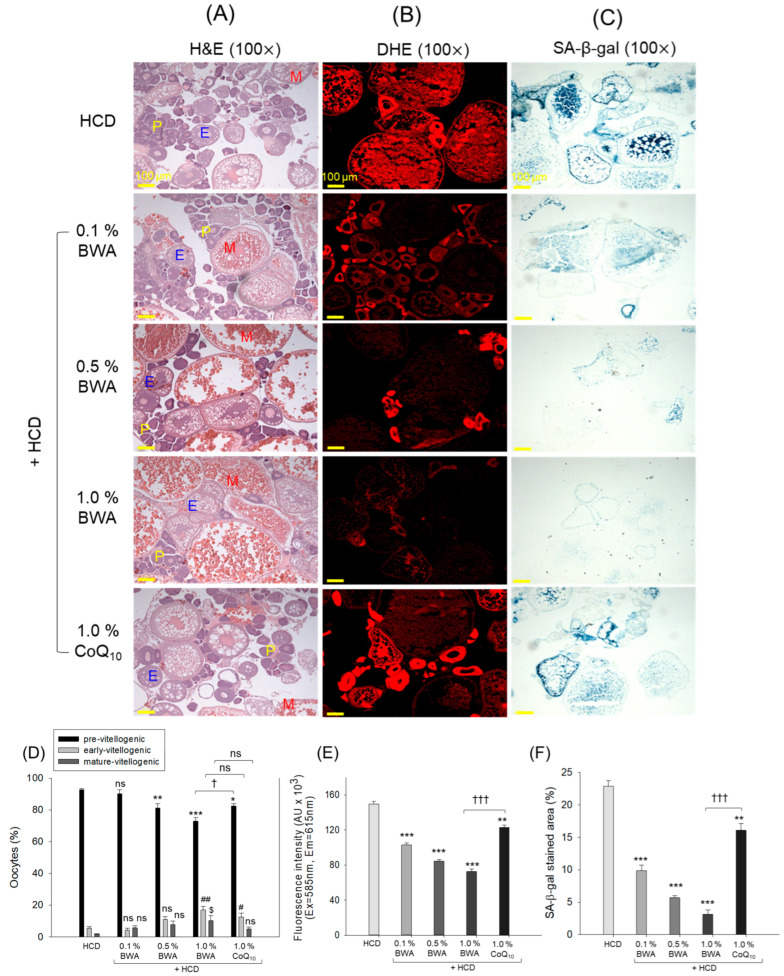
Histological examination of zebrafish ovaries following 20 weeks of consumption of different diets. (**A**) Hematoxylin and eosin (H&E) staining: premature, early and mature vitellogenic oocytes are represented by P, E and M, respectively. (**B**) Dihydroethidium (DHE) fluorescent staining for detecting reactive oxygen species (ROS). (**C**) Senescence-associated β-galactosidase (SA-β-gal) staining [scale bar, 100 μm]. (**D**) Different oocyte population quantification. (**E**) Quantification of DHE fluorescent intensity by employing Image J software. (**F**) Quantification of SA-β-gal-stained area. HCD stands for the high-cholesterol diet, while BWA and CoQ_10_ stand for beeswax alcohol and coenzyme Q_10_, respectively. * (*p* < 0.05), ** (*p* < 0.01) and *** (*p* < 0.001) depict the level of significance for the previtellogenic oocytes, DHE and SA-β-gal-stained area, while ^#^ (*p* < 0.05) and ^##^ (*p* < 0.01) depict the significance level for early-vitellogenic oocytes with respect to the HCD group. ^†^ (*p* < 0.05), ^†††^ (*p* < 0.001) highlights the significance between the 1.0% BWA and 1.0% CoQ_10_ groups; the non-significant differences between the groups are represented by “ns”.

**Figure 10 pharmaceuticals-17-01434-f010:**
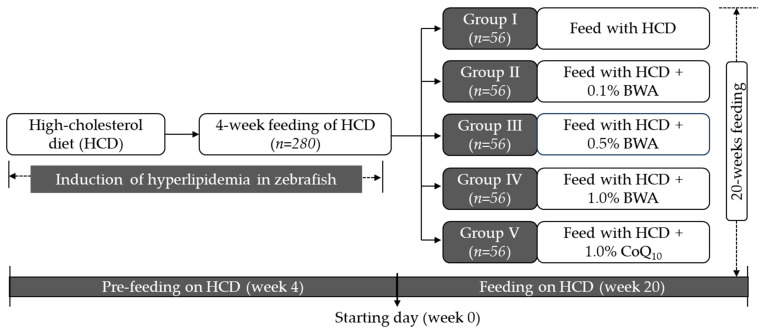
A layout of the experimental plan of the 20-week consumption study. HCD stands for the high-cholesterol diet, while BWA and CoQ_10_ stand for beeswax alcohol and coenzyme Q_10_.

**Table 1 pharmaceuticals-17-01434-t001:** Body weight (BW) of zebrafish among the different groups at the beginning and after 20 weeks supplementation of respective diets.

Diet (Total 10 mg)	HCD Alone (n = 56)	HCD+ +0.1% BWA (n = 56)	HCD +0.5% BWA (n = 56)	HCD +1.0% BWA (n = 56)	HCD +1.0% CoQ_10_ (n = 56)
BW at week 0 (mg)	421.8 ± 15.8	424.3 ± 24.2	413.5 ± 15.4	421.9 ± 25.5	429.4 ± 16.9
BW at week 20 (mg)	740.9 ± 23.3	648.7 ± 28.4	646.3 ± 22.2	688.1 ± 26.8	696.0 ± 18.1
Net increase in BW (mg)	319.1 ± 6.7	224.4 ± 9.5	232.8 ± 6.3	266.2 ± 8.9	266.6 ± 5.2
Net increase in BW (%)	175.6 ± 5.5	153.8 ± 6.7 ^ns^	152.3 ± 5.2 ^ns^	166.4 ± 6.5 ^ns^	165.0 ± 4.3 ^ns^

BW, body weight; BWA, beeswax alcohol; HCD, high-cholesterol diet; CoQ_10_, coenzymeQ_10_. ^ns^ represents the non-significant weight change in different groups at 20 weeks, compared with the HCD-only group.

**Table 2 pharmaceuticals-17-01434-t002:** Composition of different zebrafish diets prepared by mixing cholesterol, beeswax alcohol and coenzyme Q_10_ (CoQ_10_).

Diet (Total: 10 mg)	HCD Only	HCD+ +0.1% BWA	HCD +0.5% BWA	HCD +1.0% BWA	HCD +1.0% CoQ_10_
Tetrabit	9.6	9.59	9.55	9.5	9.5
Cholesterol	0.4	0.4	0.4	0.4	0.4
BWA	0	0.01	0.5	0.1	0
CoQ_10_	0	0	0	0	0.1
Total (mg)	10	10	10	10	10

BWA, beeswax alcohol; HCD, high-cholesterol diet; CoQ_10_, coenzyme Q_10_.

## Data Availability

The data used to support the findings of this study are available from the corresponding author upon reasonable request.

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
