# Peer review of "Twenty-Week Dietary Supplementation with Beeswax Alcohol (BWA; Raydel^®^) Ameliorates High-Cholesterol-Induced Long-Term Dyslipidemia and Organ Damage in Hyperlipidemic Zebrafish in a Dose-Dependent Manner: A Comparative Analysis Between BWA and Coenzyme Q_10"

_pharmaceuticals, 2024, doi:10.3390/ph17111434_

Round 1
Reviewer 1 Report
Comments and Suggestions for Authors
Dear Author,
The manuscript is an interesting and generally well written and executed study suitable for publication. However, before a final decision I would like to address some questions and suggestions; I hope you will find them useful.
First of all, Beeswax alcohol is already known for its lipotropic effect, this is why in the introduction I would highlight the novelty of the actual study. Additionally, please explain why Q vitamin was used as positive control? It is not classic medicine used to mitigate hypercholesterolemia… Similar mechanism of action? From the introduction the aim of the study is not clear for me.
Methods
4.2 Animal model… Please indicate the origin of zebrafish. Gender, age? Please mention clearly if the actual study had an approval of a bioethics committee. This is mentioned clearly in the end statement, but not in the animal chapter. In the actual form the manuscript it is mentioned that the living conditions of the animals followed the guidelines of the bioethics committee. That might cause confusion.
4.3. Please provide more details about the modified diet… was it commercial one? Was it made in the laboratory… for example how was the mixing with cholesterol done? How was assured a homogenous mixing? What devices have been used? What quality control was done…
How have been established the dose of BWA? Any preliminary studies?
I would like to know more details about the experimental protocol… how many fishes were in tank? Were they feed as a group? How do you assure that every fish gets 10 mg of the diet. Fed together, the experimental unit is the fish tank not the fish…
Please indicate how often the body weight was measured… in the methods is mentioned only “regularly”.
About killing and collection of blood and organs… Notably, in the methods, the animals were killed first… than the blood collected… Please provide some clarification on it.
Results
Please mention if they were any statistically significant differences in bw among groups. ANOVA two way? If not, I would not claim any differences if they were not statistically covered.
In my opinion whiteout a Kaplan Mayer test nothing can be sad about survival. I suggest presenting the results only, but no conclusions can be done whiteout a statistical support. Personally, I appreciate the fact that the death animals are reported, this suggests scientific correctness, which is rarely found in animal studies, but it is just not enough to say something about survival.
Fig 3 I do not understand why the title of the graph is up on the figure, when it is also written on the vertical bar of the graph and in the legend…
The other results and discussions are OK in my point of view.
Author Response
Thank you for your insightful comments. Following the reviewer’s suggestion, we made point-to-point response and reflected on revision.
Please find attached doc as our response.

Reviewer 2 Report
Comments and Suggestions for Authors
This study in the zebrafish model provides evidence that dietary supplementation of BWA induces a dose-related protection against the adverse effects of a high-fat diet on lipemic parameters and main target organs (liver, kidney, testis). The behavioural and blood assays appear convincing, whereas some of the histological data are jeopardized by major methodological flaws, likely resulting in wrong results and misleading information.
Major points
Histological analysis, liver and testis: additional details are needed to explain how morphometry was carried out on the H&E-stained sections, particularly which tissue details are included in the measured red-converted areas (a threshold method, I guess); at a glance, it seems to correspond to a faintly eosinophilic eosinophilic extracellular matrix. If so, please provide a rationale to explain why the interstitial spaces were analyzed.
Oil red O staining, all organs studied: the ORO staining lacks specificity. Being a lysochrome, ORO should only stain the lipid droplets, as those shown in Figure 4, column A (H&E), whereas it appears to aspecifically stain the whole tissue, including the blood vessel walls. Such aspecific staining conceivably results from paraffin inclusion of the tissue samples, since this histological procedure is known to dissolve and remove tissue lipids. Since this experiment adds little to the study's take-home message, it should be removed.
Acridine orange staining (all organs studied): I must also raise criticism about the lack of specificity of the acridine orange fluorescent staining used to detect apoptosis. The photographs in figure 5(A) show a diffuse, non-nuclear staining, whereas it is known that AO emits a greenish fluorescence only when it binds to DNA. The choice of using AO to detect apoptotic cells is questionable, because the apoptotic nuclei undergo DNA fragmentation which may interfere with AO-DNA binding, with obvious specificity issues. In different experimental settings, AO can label lysosomes in green, but this is not an index for apoptosis. Other methods are commonly used to detect apoptosis reliably, e.g. the TUNEL assay. If the authors do not wish to use such methods, they should at least remove the AO experiments from this manuscript.
Histological analysis, kidney: the (amorphous) basophilic structures in the upper left panel are not newly forming nephrons.
Minor points
Figure 4(D), IL-6 immunohistochemistry: please adjust the cyan background of the panel corresponding to 1% BWA to make it similar to the others.
Figure 5: some panels are out-of-focus and should be replaced
Figure 5, legend: DHE is erroneously indicated as a fluorescent dye for apoptosis, not a superoxide probe:, please check and fix.
Line 302 ‘postal’?
Comments on the Quality of English Languageminor editing of English language is suggested
Author Response

(The authors gave the same response as above.)

Round 2
Reviewer 1 Report
Comments and Suggestions for Authors
Dear Author,
I consider that almost all my request and suggestion were fulfilled. Only one issue remains… it Is totally unusual to work on laboratory animals purchased form local market… This way the health status, genetic background, age, life conditions are largely unknown… and it is illegal in some part of the world… I considered a major drawback of the study… But I fully realize that this cannot be corrected in this manuscript, this is why I recommend the publication in the actual version.
Reviewer 2 Report
Comments and Suggestions for Authors
The authors did excellent work revising the original version. All the ambiguous findings have been removed or explained better, and the histological data now appear more consistent with the functional ones.